# Cytochalasans from the Endophytic Fungus *Phomopsis* sp. shj2 and Their Antimigratory Activities

**DOI:** 10.3390/jof8050543

**Published:** 2022-05-23

**Authors:** Bing-Chao Yan, Wei-Guang Wang, Ling-Mei Kong, Jian-Wei Tang, Xue Du, Yan Li, Pema-Tenzin Puno

**Affiliations:** 1State Key Laboratory of Phytochemistry and Plant Resources in West China, Kunming Institute of Botany, Chinese Academy of Sciences, Kunming 650201, China; yanbingchao@mail.kib.ac.cn (B.-C.Y.); wwg@live.cn (W.-G.W.); konglingmei@mail.kib.ac.cn (L.-M.K.); tangjianwei@mail.kib.ac.cn (J.-W.T.); duxue@mail.kib.ac.cn (X.D.); liyan@mail.kib.ac.cn (Y.L.); 2Yunnan Key Laboratory of Natural Medicinal Chemistry, Kunming 650201, China; 3University of Chinese Academy of Sciences, Beijing 100049, China

**Keywords:** endophytic fungus, *Phomopsis*, cytochalasan, antimigratory activity

## Abstract

Cytochalasans from the endophytic fungi featured structure diversity. Our previous study has disclosed that cytochalasans from the endophytic fungus *Phomopsis* sp. shj2 exhibited an antimigratory effect. Further chemical investigation on *Phomopsis* sp. shj2 has led to the discovery of seven new cytochalasans (**1**–**7**), together with four known ones. Their structures were elucidated through extensive spectroscopic data interpretation and single-crystal X-ray diffraction analysis. Compounds **1**–**3** and **8**–**11** exhibited antimigratory effects against MDA-MB-231 in vitro with IC_50_ values in the range of 1.01−10.42 μM.

## 1. Introduction

Endophytic fungi are emerging as rich resources for structurally unique and bioactive secondary metabolites, which arouse increasing research interest in the past decades [1,2,3]. Cytochalasans represent a large class of fungal polyketide synthase-nonribosomal peptide synthetase (PKS-NRPS) hybrid secondary metabolites. Recently, plenty of polycyclic cytochalasans have been identified [4,5,6,7,8] and synthesised [9,10]; moreover, they exhibited a broad spectrum of interesting biological activities, such as cytotoxic [4,5,8], immunoregulatory [11], and antimicrobial [6] activities. To date, more than 400 cytochalasans have been isolated from various fungal sources, such as *Phomopsis* [4], *Xylaria* [12], *Chaetomium* [13] and *Phoma* [14] genera.

Tumour spread is a major concern in cancer therapeutics as cancer metastasis is responsible for 90% of deaths from solid tumours [15]. Natural products with antimigratory activity represent a highly interesting field to explore for cancer chemoprevention and therapy. Fungi are emerging as a natural source, such as *Diaporthe* [16], *Isaria* [17], and *Phenicillium* [18,19] genera. Chemical investigations on endophytes of *Isodon* species have disclosed structurally diverse and bioactive natural products [19,20,21,22]. Phomopchalasins B and C were isolated from the endophytic fungus *Phomopsis* sp. shj2 from the stems of *Isodon eriocalyx* var. *laxiflora* and exhibited in vitro antimigratory effects against MDA-MB231 [19]. In our continuous efforts for more bioactive structures, the strain was further investigated by one strain-many compounds strategy (OSMAC), which led to the isolation of seven new cytochalasans (**1**–**7**), along with four known ones (Figure 1). Herein, we report the isolation, structure elucidation, and antimigratory activities of these cytochalasans.

## 2. Materials and Methods

### 2.1. General Experimental Procedures

Column chromatography (CC) was performed with silica gel (100–200 mesh, Qingdao Marine Chemical, Inc., Qingdao, China), Lichroprep RP-18 gel (40–63 μm, Merck, Darmstadt, Germany). Preparative HPLC and semi-preparative HPLC were performed on an Agilent 1200 liquid chromatograph with a Zorbax SB-C18 (9.4 mm × 25 cm) column. Fractions were monitored by TLC, and spots were visualized by heating silica gel plates sprayed with 10% H_2_SO_4_ in EtOH. Petroleum ether (PE, 60–90 °C), EtOAc, CHCl_3_, acetone, MeOH, and EtOH were of analytical grade and purchased from Sinopharm Chemical Reagent Co. Ltd., China. All solvents were distilled before use. NMR spectra were recorded on Bruker DRX-500, AV-600, and 800 spectrometers. ESIMS and HRESIMS experiments were performed on a Bruker HCT/Esquire spectrometer and a Waters AutoSpec Premier P776 spectrometer. CD spectra were measured on an Applied Photophysics Chirascan spectrophotometer. Optical rotations were measured with a JASCO P-1020 polarimeter. UV spectra were obtained using a Shimadzu UV-2401A spectrophotometer.

### 2.2. Fungal Material

The culture of *Phomopsis* sp. shj2 was isolated from the stems of *Isodon eriocalyx* var. *laxiflora* collected from Kunming Botanical Garden, Kunming, People’s Republic of China, in December 2012. The isolate was identified based on sequence (GenBank Accession No. KU533636) analysis of the ITS region of the rDNA. The fungal strain was cultured on slants of potato dextrose agar at 25 °C for 7 days. Agar plugs were cut into small pieces (about 0.5 × 0.5 × 0.5 cm^3^) under aseptic conditions, and 15 pieces were used to inoculate three Erlenmeyer flasks (250 mL), each containing 50 mL of media (0.4% glucose, 1% malt extract, and 0.4% yeast extract); the final pH of the media was adjusted to 6.5, and the flasks were sterilized by autoclave. Three flasks of the inoculated media were incubated at 28 °C on a rotary shaker at 180 rpm for 5 days to prepare the seed culture. Fermentation was carried out in 125 Fernbach flasks (500 mL), each containing 80 g of rice. Spore inoculum was prepared in sterile, distilled H_2_O to give a final spore/cell suspension of 1 × 10^6^/mL. Distilled H_2_O (120 mL) was added to each flask, and the contents were soaked overnight before autoclaving at 15 psi for 30 min. After cooling to room temperature, each flask was inoculated with 5.0 mL of the spore inoculum and incubated at 28 °C for 42 days.

### 2.3. Extraction and Isolation

The fermented material was extracted with EtOAc (4 × 10.0 L) and the organic solvent was evaporated to dryness under vacuum to afford a crude extract (170 g). The crude extract was purified by CC (column chromatography on SiO_2_ with CHCl_3_/acetone gradient system 1:0, 9:1, 8:2, 7:3, 6:4 and 1:1) to yield six main fractions, Fr.s A–F. Fr. B was subjected to chromatography over silica gel CC (petroleum ether-EtOAc) to give subfractions B1–B9. Fr. B2 was further purified by silica gel CC (petroleum ether-acetone) to give **1** (10.7 mg). Fr. B8 was purified by semi-preparative HPLC (3 mL/min, detector UV λ_max_ 210 nm, MeCN-H_2_O) to afford **11** (3.2 mg), **8** (25.1 mg), and **10** (3.7 mg). Fr. C was purified by chromatography over silica gel CC (petroleum ether-acetone) to give subfractions Fr.s C1–C10. The subfraction C8 was recrystallized to give **7** (20.5 mg). Fr. C5 was separated by semi-preparative HPLC (3 mL/min, detector UV λ_max_ 210 nm, MeCN-H_2_O) to afford **3** (4.7 mg) and **9** (10.2 mg). Fr. D was subjected to Sephadex LH-20 (CH_3_Cl-MeOH) to yield subfractions D1–D6. The subfraction D5 was purified by recrystallization to afford **4** (1.2 mg). And Fr. D5 was further purified to afford **2** (20.3 mg). Fr. E was purified by semi-preparative HPLC (3 mL/min, detector UV λ_max_ 210 nm, MeCN-H_2_O) to afford **5** (1.5 mg) and **6** (1.6 mg).

18-Acetoxycytochalasin H (**1**): white powder (MeOH); ${\left[\alpha \right]}_{D}^{20}$ = +44.2 (c 0.23, MeOH), UV (MeOH) λ_max_ (log ε): 203.2 (0.5151); ^1^H and ^13^C NMR data, see Table 1 and Table 2; HRESIMS [M + Na]^+^ *m*/*z* 558.2826 (calcd for C_32_H_41_NO_6_Na, 558.2826).

18-Ethoxycytochalasin H (**2**): white solid; ${\left[\alpha \right]}_{D}^{18}$ = +39.0 (*c* 0.15, MeOH), UV (MeOH) λ_max_ (log ε): 204.0 (0.4717); ^1^H and ^13^C NMR data, see Table 1 and Table 2; HRESIMS [M + Na]^+^
*m*/*z* 544.3030 (calcd for C_32_H_43_NO_5_Na, 544.3039).

18-Acetoxycytochalasin J (**3**): ${\left[\alpha \right]}_{D}^{22}$ = +22.0 (*c* 0.24, MeOH), UV (MeOH) λ_max_ (log ε): 204.0 (0.5467); ^1^H and ^13^C NMR data, see Table 1 and Table 2; HRESIMS [M + Na]^+^
*m*/*z* 516.2726 (calcd for C_30_H_39_NO_5_Na, 516.2720).

18-Ethoxycytochalasin J (**4**): ${\left[\alpha \right]}_{D}^{18}$ = +50.0 (*c* 0.19, MeOH), UV (MeOH) λ_max_ (log ε): 203.8 (0.5179); ^1^H and ^13^C NMR data, see Table 1 and Table 2; HRESIMS [M + H]^+^
*m*/*z* 480.3111 (calcd for C_30_H_42_NO_4_, 480.3108).

7-Oxocytochalasin H (**5**): ${\left[\alpha \right]}_{D}^{24}\;$= −12.3 (*c* 0.19, MeOH), UV (MeOH) λ_max_ (log ε): 205.0 (0.5284); ^1^H and ^13^C NMR data, see Table 1 and Table 2; HRESIMS [M + Na]^+^
*m*/*z* 514.2559 (calcd for C_30_H_37_NO_5_Na, 514.2564).

Cytochalasin H_3_ (**6**): ${\left[\alpha \right]}_{D}^{24}\;$= −63.2 (*c* 0.19, MeOH), UV (MeOH) λ_max_ (log ε): 205.5 (0.5241); ^1^H and ^13^C NMR data, see Table 1 and Table 2; HRESIMS [M + Na]^+^
*m*/*z* 516.2719 (calcd for C_30_H_39_NO_5_Na, 516.2720).

Cytochalasin H_4_ (**7**): white solid; ${\left[\alpha \right]}_{D}^{20}$ = −28.4 (*c* 0.18, MeOH), UV (MeOH) λ_max_ (log ε): 203.2 (0.5250); IR (KBr) λ_max_ 3471, 2958, 2925, 1740, 1688, 1640, 1454, 1441, 1384, 1232 cm^−1^; ^1^H and ^13^C NMR data, see Table 1 and Table 2; HRESIMS [M + H]^+^
*m*/*z* 536.3014 (calcd for C_32_H_42_NO_6_, 536.3007).

### 2.4. X-ray Crystal Structure Analysis

The intensity data for **1** and **3** were collected on a Bruker APEX DUO diffractometer using graphite-monochromated Cu Kα radiation. The structures of these compounds were solved by direct methods (SHELXS97), expanded using difference Fourier techniques, and refined by the program and full-matrix least-squares calculations. The non-hydrogen atoms were refined anisotropically, and hydrogen atoms were fixed at calculated positions. Crystallographic data for the structures of **1** (deposition number CCDC 2169670) and **3** (deposition number CCDC 2169671) have been deposited in the Cambridge Crystallographic Data Centre database. Copies of the data can be obtained free of charge from the CCDC at www.ccdc.cam.ac.uk (accessed on 1 May 2022).

Crystal data for **1**: C_32_H_41_NO_6_, M = 535.66, orthorhombic, a = 9.5019 (6) Å, b = 15.8046 (9) *Å*, c = 19.6446 (11) *Å*, α = 90.00°, β = 90.00°, γ = 90.00°, V = 2950.1 (3) *Å*^3^, T = 100 (2) K, space group P212121, Z = 4, μ (CuKα) = 0.664 mm^−1^, 11817 reflections measured, 4764 independent reflections (R_int_ = 0.0569). The final R_1_ values were 0.0917 (I > 2σ (I)). The final wR (F^2^) values were 0.2549 (I > 2σ (I)). The final R_1_ values were 0.0937 (all data). The final wR (F^2^) values were 0.2565 (all data). The goodness of fit on F^2^ was 1.163. Flack parameter = 0.3 (5). The Hooft parameter is 0.15 (11) for 1883 Bijvoet pairs. 

Crystal data for **3**: C_30_H_39_NO_5_·H_2_O, M = 511.64, monoclinic, a = 9.7873 (3) *Å*, b = 9.4430 (3) *Å*, c = 15.5029 (4) *Å*, α = 90.00°, β = 103.6560 (10)°, γ = 90.00°, V = 1392.30 (7) *Å*^3^, T = 100 (2) K, space group P21, Z = 2, μ (CuKα) = 0.678 mm^−1^, 9344 reflections measured, 3802 independent reflections (R_int_ = 0.0470). The final R_1_ values were 0.0600 (I > 2σ (I)). The final wR (F^2^) values were 0.1736 (I > 2σ (I)). The final R_1_ values were 0.0681 (all data). The final wR (F^2^) values were 0.2047 (all data). The goodness of fit on F^2^ was 1.093. Flack parameter = 0.1 (3). The Hooft parameter is 0.31 (9) for 1240 Bijvoet pairs.

### 2.5. Antimigration Assay

Cell migration was determined using the Oris™ Pro Cell Migration Assay (Platypus Technologies, Madison, WI, USA), according to the manufacturer’s protocol. Briefly, MDA-MB-231 cells were seeded and incubated (37 °C, 5% CO_2_) for 1 h, and then indicated concentrations of compounds were added and incubated with cells for an additional 18 h. At the end of incubation, the cell viability was evaluated with MTS assays and the migration area of each group was calculated and analysed, and the results of each subgroup were presented as a percentage of DMSO-treated cells.

## 3. Results and Discussion

### 3.1. Structure Elucidation

The molecular formula of 18-acetoxycytochalasin H (**1**) was determined to be C_32_H_41_NO_6_ on the basis of HRESIMS ion at *m*/*z* 558.2826 [M + Na]^+^ (calcd. 558.2826), indicating 13 degrees of unsaturation. Its ^1^H NMR data (Table 1) showed typical signals of three tertiary methyl groups (*δ*_H_ 2.24, s; *δ*_H_ 2.00, s; *δ*_H_ 1.58, s), two secondary methyl groups (*δ*_H_ 0.99, d, *J* = 6.7 Hz; *δ*_H_ 1.02, d, *J* = 6.9 Hz), six olefinic protons (*δ*_H_ 5.85, dd, *J* = 16.6, 2.3 Hz; *δ*_H_ 5.74, dd, *J* = 15.5, 9.7 Hz; *δ*_H_ 5.56, d, *J* = 16.6 Hz; *δ*_H_ 5.38, m; *δ*_H_ 5.33, s; *δ*_H_ 5.10, s), two oxygenated methine groups (*δ*_H_ 5.63, d, *J* = 2.3 Hz; *δ*_H_ 3.84, d, *J* = 10.5 Hz), and one single-substituted phenyl (*δ*_H_ 7.31, t, *J* = 7.4 Hz, 2H; *δ*_H_ 7.25, t, *J* = 7.4 Hz, 1H; *δ*_H_ 7.14, d, *J* = 7.4 Hz, 2H). The ^13^C NMR data (Table 2) displayed resonances for 32 carbons, ascribed to 5 methyls, 4 methylenes (including 1 olefinic), 11 methines (4 olefinic and 2 oxygenated), 61 quaternary carbons (1 olefinic, 1 amide and 2 ester carbonyls), and 6 other signals assignable to the single-substituted phenyl group. Thus, the above-mentioned results indicated that **1** should be a new tetracyclic cytochalasin including a benzene ring, with structural similarity with cytochalasin H [23]. The manifest difference of the structure of **1** from that of cytochalasin H was an additional acetoxy group linked at C-18 (*δ*_C_ 84.4) in **1**, which was further supported by the HMBC correlation from OAc (*δ*_H_ 2.24, s) to C-18. And the planar structure of **1** was established by extensive analysis of its 2D NMR spectra (Figure 2); its relative configuration was determined by the ROESY correlations (Figure 3) and comparative analysis of those of cytochalasin H. Fortunately, suitable crystals of **1** were obtained and subjected to X-ray diffraction analysis using Cu K*α* radiation (Figure 4), which confirmed the above deductions and unambiguously determined the absolute configuration of **1** as *3S,**4R,5S,7S,8R,9R,16S,18R,21R* with the Hooft parameter 0.15 (11) for 1883 Bijvoet pairs (CCDC 2169670).

18-Ethoxycytochalasin H (**2**) was obtained as a white powder; its molecular formula was established as C_32_H_43_NO_5_ on the basis of the HRESIMS ion peak at *m*/*z* 544.3030 [M + Na]^+^ (calcd for C_32_H_43_NO_5_Na, 544.3039), indicating 12 degrees of unsaturation. Analyses of the NMR data of **2** with those of **1** indicated their structural similarities, except for an ethoxy group located at C-18 in **2** rather than the 18-OAc group in **1**, which was confirmed by the ^1^H-^1^H COSY correlation of CH_2_ (*δ*_H_ 3.38, m; 2.65, m)/CH_3_ (1.14, t, *J =* 6.9 Hz) in the ethoxy group and the HMBC correlations from CH_2_-18-OEt (*δ*_H_ 3.38, m; 2.65, m) to C-18 (*δ*_C_ 78.5) (Figure 2). The relative configurations of C-3, C-4, C-5, C-7, and C-8 in **2** were determined to be the same as those of **1** by analysis of the ROESY spectrum (Figure 3). Considering the almost complete consistent CD spectra of **1** and **2** (see Appendix A), the absolute configuration of **2** was determined as shown.

18-Acetoxycytochalasin J (**3**) had the molecular formula of C_30_H_39_NO_5_ based on the positive HRESIMS at *m*/*z* 516.2726 [M + Na]^+^ (calcd 516.2720), corresponding to 12 degrees of unsaturation. The 1D NMR data (Table 1 and Table 2) of **3** were similar to those of cytochalasin J [24], except for an additional acetoxy group located at C-18 in **3**. The above deduction was further confirmed by the changed chemical shift of C-18, compared with the ^13^C NMR data of cytochalasin J [24], and the HMBC correlation from CH_3_-18-OAc (*δ*_H_ 1.96, s) to 18-OAc carbonyl (*δ*_C_ 170.4) (Figure 2); its structure including the relative configuration was finally established as shown by X-ray diffraction analysis (Figure 4). Considering the similar CD spectra of **1** and **3** (SI), the absolute configuration of **3** was determined to be *3S,**4R,5S,7S,8R,9R,16S,18R,21R*.

18-Ethoxycytochalasin J (**4**) had the molecular formula of C_30_H_41_NO_4_ on the basis of the positive HRESIMS (*m*/*z* 480.3111 [M + H]^+^, calcd 480.3108), corresponding to 11 degrees of unsaturation. Careful comparison of the ^1^H and ^13^C NMR spectra of **4** and **3** (Table 1 and Table 2) suggested their similar structures, except for an ethoxy group located at C-18 in **4** rather than an acetoxy group in **3**. The above deduction was supported by the ^1^H-^1^H COSY correlations of CH_2_/CH_3_ (18-OEt) and the HMBC correlation from CH_2_-18-OEt (*δ*_H_ 3.41, m; 3.37, m) to C-18 (*δ*_C_ 78.5) (Figure 2). The absolute configuration of **4** was determined to be the same as that of **1** by analysis of their CD spectra; thus, the structure of **4** was established as shown (Figure 1).

7-Oxocytochalasin H (**5**) possessed the molecular formula of C_30_H_37_NO_5_ with 13 degrees of unsaturation, which was determined by the positive HRESIMS (*m*/*z* 514.2559 [M + Na]^+^, calcd 514.2564). Analysis of the ^1^H and ^13^ C NMR data (Table 1 and Table 2) of **5** and cytochalasin H [23] indicated their structural similarity. The manifest differences were that the C-7 oxymethine group in cytochalasin H was replaced by the C-7 carbonyl group (*δ*_C_ 198.7) in **5**. The HMBC correlations from H-8 (*δ*_H_ 3.94, d, *J* = 9.3 Hz) and H_2_-12 (*δ*_H_ 6.25, s) to C-7 and other correlations in the 2D spectra of **5** confirmed the above deduction (Figure 2). The correlations of H-4/H-8, H_2_-10/H-4, and H_3_-11/H-3 in the ROESY spectrum indicated that H-3 was *a*-oriented and H-4, H-5, H-8 were *β*-oriented (Figure 3). Considering the same biogenetic pathway of **1** and **5**, the structure of **5** was determined as shown (Figure 1).

Cytochalasin H_3_ (**6**) had the molecular formula of C_30_H_39_NO_5_ with 12 degrees of unsaturation, which was determined by the positive HRESIMS (*m*/*z* 516.2719 [M + Na]^+^, calcd 516.2720). Detailed analysis of ^1^H and ^13^C NMR data of **6** (Table 1 and Table 2) indicated that **6** possessed a similar structure to that of **5**, except for the presence of a saturated C–C bond between C-6 (*δ*_C_ 45.8) and C-12 (*δ*_C_ 16.0) in **6** rather than a terminal double bond in **5**, which was further confirmed by the ^1^H–^1^H COSY correlation of H-6/H_3_-12 and the HMBC correlations from H_3_-12 (*δ*_H_ 1.12, d, *J* = 7.0 Hz) to C-5 (*δ*_C_ 35.7) and C-7 (*δ*_C_ 214.0) (Figure 2). The ROESY correlations of H-3/H_3_-11, H_3_-11/H_3_-12, H_2_-10/H-4, and H-4/H-8 implied the *α*-orientation of H-3 and *β*-orientations of H-4, H-5, H-6, and H-8 (Figure 3). By analysis of the similar CD spectra of **6** and **1** and biogenetic consideration, the structure of **6** was determined as shown.

The molecular formula of cytochalasin H_4_ (**7**) was deduced to be C_32_H_41_NO_6_ with 13 degrees of unsaturation based on the positive HRESIMS (*m*/*z* 536.3014 [M + H]^+^, calcd 536.3007). The ^13^C NMR data (Table 1 and Table 2 of **7** displayed resonances for 32 carbons, ascribed to five methyls, four methylenes (including one oxygenated), 11 methines (5 olefinic and one oxygenated), six quaternary carbons (one olefinic, one amide and two ester carbonyls), and 6 other signals assignable to the single-substituted phenyl group. The above-mentioned results indicated the presence of an additional acetoxy group and an oxymethine group compared to those of the known RKS-1778 (**10**) [25]. The ROESY correlations of H-3/H_3_-11, H_2_-10/H-4, and H-4/H-8 implied the *α*-orientation of H-3 and *β*-orientations of H-4, H-5, and H-8 (Figure 3). The absolute configuration of **7** was determined as shown by analysis of their similar CD spectra of **7** and **10** and biogenetic consideration.

Compounds **8**–**11** were identified as cytochalasin H (**8**) [23], cytochalasin J_1_ (**9**) [24], RKS-1778 (**10**) [25], 21-acetoxycytochalasin J_2_ (**11**) [26] on the basis of their spectroscopic features and by comparison with the published data in the literature. Biogenetically, compounds **1**–**11** might be derived from a polyketide chain (octaketide) and an amino acid building block (phenylalanine) through a number of steps involving cycloaddition, oxidation, reduction, dehydration, acetylation, ethylation and methylation [27,28].

### 3.2. Antimigratory Activity

Our previous studies have revealed that phomopchalasins B and C displayed antimigratory effects [19]. In order to explore the potential of the cytochalasans on antimigration against tumours, eight compounds in sufficient natural amounts (Table 3) were evaluated for antimigratory activities against MDA-MB-231 in vitro. As a result, **1**–**3** and **8**–**11** exhibited in vitro antimigratory effects with IC_50_ values in the range of 1.01–10.42 μM (cytochalasin D as the positive control); it suggested the activity decreased when the C-18 hydroxy group was substituted with the acetoxy, ethoxy or methoxy group (**8** vs. **1**, **2**, and **9**). When a double bond was introduced between C-17 and C-18 rather than an ethoxy or methoxy group at C-18, the activity slightly improved (**11** vs. **2** and **9**). Compound **3** displayed antimigratory activity with an IC_50_ value of 6.38 μM. The introduction of an acetoxy group at C-21 may enhance the activity (**1** vs. **3**). When the unit of a terminal double bond (C6-C12) and a hydroxy group at C-7 was replaced by a trisubstituted alkene (C12-C6-C7), the activity slightly improved (**8** vs. **10**), but the further introduction of an acetoxy group at C-12 decreased the activity (**10** vs. **7**).

## 4. Conclusions

Seven new cytochalasans (**1**–**7**), together with four known ones, cytochalasin H (**8**), cytochalasin J_1_ (**9**), RKS-1778 (**10**), and 21-acetoxycytochalasin J_2_ (**11**), were isolated from *Phomopsis* sp. shj2. Their structures were elucidated through extensive spectroscopic data interpretation and single-crystal X-ray diffraction analysis. In the present study, eight cytochalasans were evaluated for their antimigratory activity. Compounds **1**–**3** and **8**–**11** exhibited antimigratory activity against MDA-MB-231 in vitro with IC_50_ values in the range of 1.01−10.42 μM. The results will lay a foundation for further study of the structure–activity relationship for the discovery of antitumour lead compounds.

## Figures and Tables

**Figure 1 jof-08-00543-f001:**
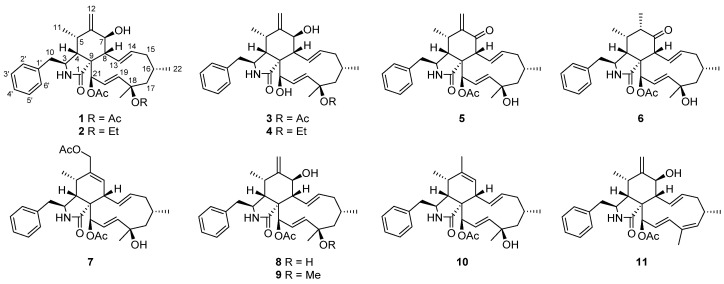
Structures of compounds **1**–**11**.

**Figure 2 jof-08-00543-f002:**
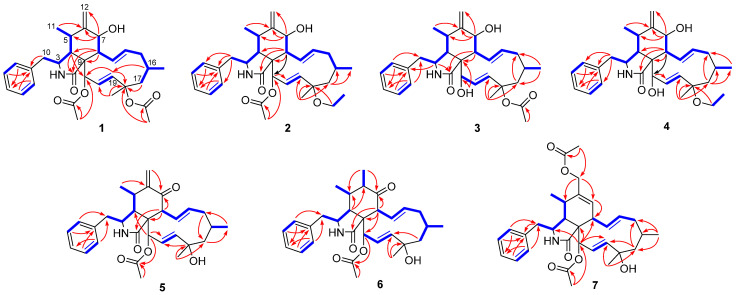
Key HMBC (red arrows) and ^1^H-^1^H COSY (blue bold) correlations of compounds **1**–**7**.

**Figure 3 jof-08-00543-f003:**
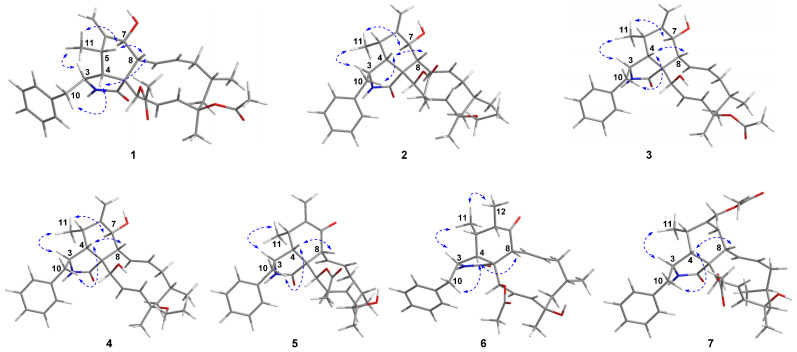
Key ROESY correlations of compounds **1**–**7**.

**Figure 4 jof-08-00543-f004:**
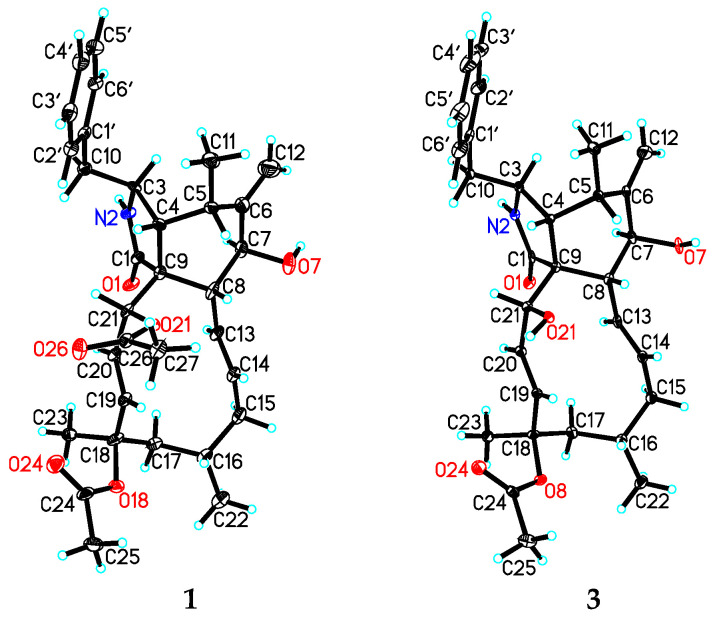
X-ray crystallographic structures of compounds **1** and **3**.

**Table 1 jof-08-00543-t001:** ^1^H NMR data (CDCl_3_, *δ* in ppm) of compounds **1**–**7**.

No.	1 ^a,b^	2 ^a,c^	3 ^c,d^	4 ^a,c^	5 ^a,c^	6 ^a,e^	7 ^a,c^
3	3.25 (m)	3.26 (m)	3.33 (m)	3.27 (m)	3.23 (dt, 9.4, 4.3)	3.54 (dt, *J* = 9.4, 4.3)	3.28 (overlap)
4	2.15 (m)	2.14 (m)	2.65 (m)	2.58 (m)	2.35 (t, 4.3)	2.25 (t, 4.2)	2.19 (t, 4.3)
5	2.76 (m)	2.78 (m)	2.72 (m)	2.90 (m)	3.08 (m)	2.11 (m)	2.53 (m)
6						2.01 (m)	
7	3.84 (d, 10.5)	3.83 (d, 10.5)	3.79 (d, 10.5)	3.82 (d, 10.5)			5.72 (s)
8	2.93 (d, 10.5)	2.96 (d, 10.5)	2.94 (d, 10.5)	2.90 (d, 10.5)	3.94 (d, 9.3)	3.79 (d, 9.4)	3.26 (overlap)
10	2.85 (dd, 13.5, 4.5) 2.65 (dd, 13.5, 9.6)	2.86 (dd, 13.3, 3.8) 2.67 (m)	2.81 (m) 2.79 (m)	2.58 (m) 1.71 (m)	2.90 (dd, 13.5, 4.3) 2.65 (dd, 13.5, 9.4)	2.92 (dd, 13.5, 4.3) 2.64 (dd, 13.5, 9.4)	2.91 (dd, 13.5, 4.3) 2.60 (dd, 13.5, 10.2)
11	0.99 (d, 6.7)	1.01 (d, 6.7)	0.84 (d, 6.8)	1.10 (d, 6.7)	1.12 (d, 6.7)	0.98 (d, 6.7)	1.17 (d, 7.3)
12	5.33 (s) 5.10 (s)	5.35 (s) 5.11 (s)	5.18 (s) 4.95 (s)	5.32 (s) 5.11 (s)	6.25 (s) 5.29 (s)	1.12 (d, 7.0)	4.53 (d, 12.8) 4.48 (d, 12.8)
13	5.74 (dd, 15.5, 9.7)	5.73 (dd, 15.1, 10.0)	5.70 (dd, 15.0, 9.2)	5.71 (dd, 15.5, 9.8)	5.81 (dd, 15.6, 9.3)	5.69 (dd, 15.5, 9.4)	5.84 (dd, 15.3, 10.3)
14	5.38 (m)	5.43 (m)	5.22 (m)	5.35 (m)	5.19 (m)	5.16 (m)	5.24 (m)
15	2.01 (overlap) 1.79 (d, 12.4)	2.00 (overlap) 1.79 (d, 11.3)	1.90 (dd, 13.9, 3.1) 1.80 (m)	1.98 (dd, 10.4, 4.7) 1.78 (m)	2.04 (dd, 12.9, 4.4) 1.89 (m)	2.01 (m) 1.81 (m)	1.99 (m) 1.77 (overlap)
16	1.65 (m)	1.78 (m)	1.66 (m)	1.75 (m)	1.76 (m)	1.75 (m)	1.75 (m)
17	2.05 (dd, 14.3, 3.7), 1.75 (dd, 14.3, 3.0	1.69 (m)	1.90 (m) 1.75 (m)	1.78 (overlap)	1.85 (overlap) 1.54 (dd, 14.3, 3.2)	1.85 (m) 1.53 (dd, 14.3, 3.1)	1.88 (dd, 14.3, 2.7) 1.54 (d, 14.3)
19	5.56 (d, 16.6)	5.52 (d, 16.7)	5.84 (d, 16.7)	5.73 (d, 16.7)	5.52 (d, 16.6)	5.49 (d, 16.6)	5.52 (d, 16.6)
20	5.85 (dd, 16.6, 2.3)	5.79 (dd, 16.7, 2.4)	5.97 (dd, 16.7, 2.2)	5.99 (dd, 16.7, 2.6)	5.90 (dd, 16.6, 2.6)	5.85 (dd, 16.6, 2.5)	5.91 (dd, 16.6, 2.6)
21	5.63 (t, 2.3)	5.54 (t, 2.4)	4.02 (t, 2.2)	4.12 (t, 2.6)	5.65 (t, 2.6)	5.60 (t, 2.5)	5.68 (t, 2.6)
22	1.02 (d, 6.9)	1.01 (d, 6.5)	0.99 (d, 6.3)	1.01 (d, 6.3)	1.04 (d, 7.0)	1.03 (d, 6.9)	1.04 (d, 6.3)
23	1.58 (s)	1.26 (s)	1.53 (s)	1.28 (s)	1.34 (s)	1.32 (s)	1.34 (s)
2′, 6′	7.14 (d, 7.4)	7.15 (d, 7.4)	7.21 (d, 7.3)	7.15 (d, 7.4)	7.12 (d, 7.4)	7.15 (d, 7.3)	7.14 (d, 7.2)
3′, 5′	7.31 (t, 7.4)	7.32 (t, 7.4)	7.29 (t, 7.3)	7.31 (t, 7.4)	7.32 (t, 7.4)	7.33 (t, 7.3)	7.31 (t, 7.5)
4′	7.25 (t, 7.4)	7.25 (t, 7.4)	7.26 (d, 7.3)	7.24 (t, 7.4)	7.25 (t, 7.4)	7.25 (t, 7.3)	7.24 (t, 7.2)
12-OAc							2.04, s
18-OR	R = Ac 2.00 (s)	R = Et 3.38 (m), 2.65 (m) 1.14 (t, 6.9)	R = Ac 1.96 (s)	R = Et 3.41 (m), 3.37 (m) 1.17 (t, 7.0)	R = H	R = H	R = H
21-OAc	2.24 (s)	2.25 (s)			2.30 (s)	2.28 (s)	2.25 (s)

^a^ Recorded in CDCl_3_. ^b^ Recorded at 800 MHz. ^c^ Recorded at 600 MHz. ^d^ Recorded in acetone-*d*_6_. ^e^ Recorded at 500 MHz.

**Table 2 jof-08-00543-t002:** ^13^C NMR data (CDCl_3_, *δ* in ppm) of compounds **1**–**7**.

No.	1 ^a,b^	2 ^a,c^	3 ^d,e^	4 ^a,e^	5 ^a,e^	6 ^a,f^	7 ^a,e^
1	174.3 s	174.3 s	176.7 s	175.8 s	172.8 s	173.5 s	174.9 s
3	53.9 d	53.9 d	54.2 d	50.6 d	54.0 d	53.3 d	56.1 d
4	50.5 d	50.9 d	50.2 d	53.9 d	50.6 d	51.1 d	53.8 d
5	33.0 d	33.0 d	33.8 d	33.1 d	34.2 d	35.7 d	34.7 d
6	148.0 s	148.0 s	152.1 s	148.6 s	143.9 s	45.8 d	135.8 s
7	70.0 d	69.9 d	71.6 d	70.1 d	198.7 s	214.0 s	134.6 d
8	47.3 d	47.4 d	46.8 d	46.0 d	53.1 d	52.0 d	43.4 d
9	52.1 s	51.9 s	54.5 s	53.0 s	52.9 s	53.6 s	56.2 s
10	45.7 t	45.8 t	45.5 t	45.8 t	46.0 t	46.3 t	46.1 t
11	14.1 q	14.2 q	14.2 q	14.1 q	14.4 q	15.9 q	13.2 q
12	114.3 t	114.3 t	112.0 t	113.9 t	121.0 t	16.0 q	64.9 t
13	127.6 d	127.2 d	130.2 d	128.0 d	123.0 d	123.2 d	128.3 d
14	138.2 d	138.6 d	135.7 d	137.8 d	138.4 d	137.9 d	136.3 d
15	42.7 t	43.1 t	43.7 t	42.8 t	43.1 t	42.9 t	42.8 t
16	28.6 d	28.0 d	29.2 d	27.8 d	28.6 d	28.5 d	28.7 d
17	51.5 t	51.8 t	52.9 t	51.1 t	53.7 t	53.5 t	53.5 t
18	84.4 s	78.5 s	85.0 s	78.5 s	74.6 s	74.5 s	74.6 s
19	136.6 d	138.9 d	135.1 d	137.3 d	137.7 d	137.7 d	137.3 d
20	124.9 d	125.8 d	131.8 d	130.8 d	125.9 d	125.9 d	126.5 d
21	77.4 d	78.1 d	76.7 d	77.0 d	77.9 d	77.8 d	77.0 d
22	25.5 q	26.2 q	25.9 q	26.0 q	26.6 q	26.6 q	26.6 q
23	26.3 q	25.2 q	27.0 q	25.2 q	31.5 q	31.4 q	31.6 q
1′	137.5 s	137.6 s	138.9 s	137.7 s	137.0 s	137.1 s	137.7 s
2′, 6′	129.1 d	129.2 d	130.8 d	129.1 d	129.2 d	129.2 d	129.1 d
3′, 5′	129.1 d	129.1 d	129.2 d	128.9 d	129.1 d	129.1 d	129.0 d
4′	127.2 d	127.3 d	127.4 d	127.1 d	127.4 d	127.4 d	127.3 d
12-OR							170.6 s, 21.1 q
18-OR	R = Ac 170.1 s, 21.0 q	R = Et 57.9 t, 16.3 q	R = Ac 170.4 s, 22.2 q	R = Et 57.5 t, 16.0 q	R = H	R = H	
21-OAc	170.3 s, 22.4 q	170.3 s, 21.1 q			170.1 s, 21.1 q	170.1 s, 21.1 q	170.2 s, 21.2 q

^a^ Recorded in CDCl_3_. ^b^ Recorded at 100 MHz. ^c^ Recorded at 150 MHz. ^d^ Recorded in acetone-*d*_6_. ^e^ Recorded at 150 MHz. ^f^ Recorded at 125 MHz.

**Table 3 jof-08-00543-t003:** Antimigratory activities of the compounds against MDA-MB-231 in vitro.

Compounds	IC_50_ (μM)	Compounds	IC_50_ (μM)
Cytochalasin D ^a^	0.78	**8**	1.25
**1**	3.14	**9**	7.31
**2**	10.42	**10**	1.01
**3**	6.38	**11**	6.41
**7**	>25		

^a^ Positive control.

## Data Availability

X-ray crystallographic data of **1** and **3** (CIF) are available free of charge from the CCDC at https://www.ccdc.cam.ac.uk (accessed on 1 May 2022).

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
