# Peer review of "Cytochalasans from the Endophytic Fungus Phomopsis sp. shj2 and Their Antimigratory Activities"

_jof, 2022, doi:10.3390/jof8050543_

Round 1
Reviewer 1 Report
Dear authors!
Text of manuscript by Bin-Chao Yan et al. are interesting, but required some corrections and improves.
Figure 1 - Superfluous stereodescriptors for C-18. Please, to depict the stereocenters using the rule: one stereocenter - one stereodescriptor. Also, please indicate the R- or S-orientation of stereocenters in this figure. And, in numbering of compound 1 there is "14" depicted twice.
Introduction: Need to add more detailed description about anti-migration activity and such activity of fungal metabolites in particular.
Fungal material. Is this your standard cultivation method or did you specifically select such conditions for this fungus?
Table 2. For compounds 1-4 and 7 you've added character of splitting, but not for compounds 5 and 6. Why? Please, bring the table to a single form.
Nowhere in the text of the article is it found which exact configurations of the stereocenters were determined. I mean, need to be written, for example: "absolute configurations were determined as 2S,10R,24S etc."
Figure 2 - Please, add the key numbers of atoms in structure of 1.
Figure 3 - Please, increase the scale of picture.
Antimigratory activity. Very meager description the results and discussion about SAR, for my opinion.
Also corrections you can found in attached.

Reviewer 2 Report
The authors presented the isolation and structural determination of seven new cytochalasans (1-7) from the endophytic fungus Phomopsis sp. Shj2. This manuscript contains interesting new findings and can be acceptable for publication in Journal of Fungi after minor revision. Please check the following points.
- Lines 94, 100, 177, and 200; Ehtyl→Ethoxy
- Lines 183, 204, 213, 223, and 232; The authors should determine on the absolute configuration of compounds 2, 4, 5, 6 and 7 using data such as specific rotation or CD spectral data of these compounds.
- Line 198; The authors should determine the absolute configuration of compound 3 from the results of X-ray diffraction analysis as in compound 1.
- The authors should comment on the conformational characteristics of compounds 1 and 3 in solution and in crystalline state.
- The authors should comment on the biosynthetic pathway of compounds 1-11 in Phomopsis sp. Shj2.
- Are the ethoxy group at C-18 in compounds 1 and 3 naturally derived? Or is there a possibility of an artifact? The authors should comment on this point.
Round 2
Reviewer 1 Report
Dear authors! I very glad to see, that you improved manuscript text according to my previous corrections and remarks.
However, Supplementary material need to be small corrected.
HRESIMS, CD and UV spectra of compounds 1-7 need to be presented in more high resolution. Also, in signature of 13C spectra should be mention the DEPT spectra, for example: "13C and DEPT-135 spectra of compound XX"
Good luck!
